# Clinicopathological Features and Survival of Patients with Hepatocellular Carcinoma in Ethiopia: A Multicenter Study

**DOI:** 10.3390/cancers15010193

**Published:** 2022-12-28

**Authors:** Getahun Befirdu Abza, Jemal Hussien Ahmed, Abdu Adem Yesufe, Edom Seife, Mengistu Erkie, Isabel Spriet, Legese Chelkeba, Pieter Annaert

**Affiliations:** 1Drug Delivery and Disposition, Department of Pharmaceutical and Pharmacological Sciences, KU Leuven, 3000 Leuven, Belgium; 2Department of Pharmacology, School of Pharmacy, Jimma University, Jimma P.O.Box 378, Ethiopia; 3Adult Oncology Unit, St. Paul’s Hospital Millennium Medical College, Addis Ababa 1000, Ethiopia; 4Department of Radiotherapy and Adult Oncology, Tikur Anbessa Specialized Hospital, Addis Ababa University, Addis Ababa 1000, Ethiopia; 5Department of Internal Medicine, Division of Gastroenterology & Hepatology, Addis Ababa University, Addis Ababa 1000, Ethiopia; 6Clinical Pharmacology and Pharmacotherapy, KU Leuven Department of Pharmaceutical and Pharmacological Sciences, 3000 Leuven, Belgium; 7Department of Pharmacology and Clinical Pharmacy, School of Pharmacy, Addis Ababa University, Addis Ababa 1000, Ethiopia

**Keywords:** hepatocellular carcinoma, hepatitis, survival outcome, prognostic factor

## Abstract

**Simple Summary:**

Hepatocellular carcinoma (HCC) is the most common form of liver cancer with a low survival rate worldwide. The number of new cases and deaths from HCC is increasing globally, thus, we investigated the clinical conditions and survival of patients in Ethiopia to understand the extent of the problem and develop prevention and control strategies. Our analysis showed that hepatitis B virus (HBV) is an important risk factor associated with HCC. Half of the patients involved in our study survived only for about 5 months after their diagnosis. Patients who had been given antiviral therapy for an HBV infection survived longer than those who were not treated. By the year 2040, there will be a two-fold increase in the number of cases and deaths from HCC in Ethiopia. Therefore, since HBV is a major underlying factor for HCC, it is crucial to increase the vaccination coverage and access to antiviral drugs against hepatitis to lessen the devastation.

**Abstract:**

(1) Background: Hepatocellular carcinoma (HCC) is one of the deadliest cancers globally, killing over 700,000 people each year. Despite the rising incidence and mortality rates of HCC in Ethiopia, only few single-centered studies have been conducted; therefore, we aimed to explore the clinicopathological characteristics and survival of patients with HCC in multicenter settings. (2) Methods: We conducted a retrospective analysis of 369 patients with confirmed HCC diagnosed between 2016 and 2021. The survival of patients weas determined using the Kaplan–Meier method, and hazard ratios of the prognostic factors were estimated in Cox proportional hazard models. (3) Results: Majority patients were male (67%) and had a mean age of 52.0 ± 15.6 years. The majority of patients (87%) had a large tumor size (>5 cm) at diagnosis and presented with an advanced-stage condition. Cirrhosis (58%) and viral hepatitis (46.5%) were the main risk factors associated with HCC. The median overall survival was 141 days (95% CI: 117–165). Patients who took antivirals for HBV had a higher survival benefit compared to the untreated group (469 vs. 104 days; *p* < 0.001). The risk of death was 12 times higher in patients with Barcelona Clinic Liver Cancer-D (BCLC-D) terminal stage HCC compared to patients with an early stage (BCLC-A) HCC. The stage of HCC and treatment against HBV are the most significant survival predictors. (4) Conclusions: The overall survival of HCC patients in Ethiopia is poor. Cirrhosis and viral hepatitis are the primary risk factors linked with HCC. Patients who received antiviral therapy for HBV had a better survival outcome.

## 1. Introduction

Hepatocellular carcinoma (HCC) is one of the deadliest cancers globally, killing more than 700,000 people each year [1]. The incidence rate of HCC in sub-Saharan Africa and in Asia is disproportionally high compared to developed nations, with over 20 people per 100,000 of the population [2]. A report indicated that HCC is one of the major causes for cancer-related deaths in sub-Saharan Africa [2]. In Ethiopia, the age-adjusted incidence and mortality rates of HCC is 2.7 and 2.7 per 100,000 inhabitants, respectively [3]. The common risk factors for HCC include non-alcoholic steatohepatitis (NASH), viral hepatitis, cirrhosis, and an excessive alcohol intake [4]. In sub-Saharan Africa, the major risk factor for HCC is hepatitis; more than half of the cases are linked to a hepatitis B virus (HBV) infection. Ethiopia is one of the HBV endemic countries, with a prevalence rate of 9.4% among the population aged 15 years and above [5].

The survival of HCC patients varies depending on the type of treatment, stages of cancer, and comorbidity status. In sub-Saharan countries including Ethiopia, only few patients (3%) receive disease-specific treatments [6], and chemotherapeutics such as 5-fluorouracil, oxaliplatin, and doxorubicin are the commonly used treatments. However, these cytotoxic agents cause severe side effects and are linked to multidrug resistance resulting in therapeutic failure. The molecular-targeted tyrosine kinase inhibitors, sorafenib and lenvatinib, are unavailable and/or unaffordable for most Ethiopians who have an average monthly income of ~200 USD [7]. For instance, sorafenib costs about 215–350 USD per month (E. Amare and N. Endeshaw, personal communications, June 8, 2022). Moreover, HCC is usually presented in the advanced- or end-stage status, which leads to a poor clinical prognosis, explaining why patients survived only for few months after their diagnosis [8]. In Ethiopia, an increased incidence of HCC-related morbidity and mortality is also associated with a rising burden of viral hepatitis [9]. The high fatality rate highlights the inadequacy of effective treatments as well as a lack of public awareness regarding the risk factors [4,9]. Despite the high incidence of HCC in Ethiopia, the clinical and pathological characteristics and survival outcomes have not been well explored. Therefore, this study aimed to assess the clinicopathological features, management, and survival outcome of HCC in Ethiopia. To the best of our knowledge, this is the first multicenter study focused on Ethiopian HCC patients.

## 2. Materials and Methods

### 2.1. Study Design and Population

A multicenter retrospective study was conducted at four referral oncology care centers: Tikur Anbessa Specialized Hospital, Saint Paul’s Hospital Millennium Medical College, Hawassa University Comprehensive Specialized Hospital, and Jimma University Medical Center. Patients with confirmed HCC diagnosed between 2016 and 2021 were enrolled in our study. A total of 694 patients were identified, of which 369 patients fulfilled the set criteria and were included in the analysis (Figure 1). Patients with primary HCC whose age was 18 years and above were included. The number of patients’ records reviewed among the study sites varied depending on their service years and capacity.

### 2.2. Methods

All the data were collected using a pre-tested data extraction format. We assessed the medical records of the patients to confirm their diagnosis. Patients were diagnosed clinically with a combination of radiological imaging techniques, histopathology, and serological markers. The radiological diagnosis of HCC was conducted mainly with the triphasic abdominal CT according to the guidelines of the National Comprehensive Cancer Network (NCCN) [10]. Depending on the resources available in oncology centers, liver ultrasound imaging was also used to further characterize the HCC lesions. When the imaging results were uncertain, histological examinations were conducted, mainly on HCC patients without cirrhosis. Cirrhotic patients were diagnosed primarily with ultrasound imaging. Later on, in consultation with medical oncologists, we set an operational definition for cirrhosis (i.e., *patients who presented with ascites, portal vein thrombosis, jaundice, and hepatic encephalopathy were considered as cirrhotic patients even though they were not labelled as such on their medical chart*) to further assess the symptoms and imaging records and categorize the patients as cirrhotic and non-cirrhotic. Furthermore, patients with cirrhosis were classified as compensated and decompensated depending on their symptomatic features. Compensated cirrhosis was defined as uncomplicated and is an asymptomatic type of cirrhosis that can be detected with an incidental finding of elevated liver enzymes and other serum biomarkers [11]. In contrast, patients presenting with ascites, jaundice, and hepatic encephalopathy were considered as having decompensated cirrhosis [11]. Baseline laboratory indices such as the alpha feto-protein (AFP), aminotransferases, albumin, and bilirubin levels were recorded. The potential risk factors and clinicopathological data, such as the size and type of the tumors, the presence of metastasis, a history of cirrhosis, and the patients’ baseline performance, as scored by the Eastern Cooperative Oncology Group (ECOG) system, were reviewed and collected. The stages of HCC were stratified by the Barcelona Clinic Liver Cancer (BCLC) staging system (Appendix A). Please note that patients with ECOG > 2 were classified as a terminal (BCLC-D) stage HCC, as outlined in the latest version of the NCCNs guidelines [10]. Information such as the types of treatment given to HCC patients were collected on the basis of the available data. In most oncology centers, the management of HCC was decided by a multidisciplinary team consisting of oncologists, hepatologists, a hepatobiliary surgeon, and radiologists (pathologists as well). The first date of their diagnosis and the patients’ last follow-up were essential information in determining the survival time. In some cases, we confirmed whether the patients were dead or alive by making phone calls to families and/or friends. 

### 2.3. Outcome Measures

The primary outcome was the overall survival (OS) of HCC patients, which is defined as the time between the patients’ diagnosis and death from any cause. Patients were censored if they lost from follow-up or were alive at the end of the study.

### 2.4. Statistical Analysis

All the statistical analyses were conducted using SPSS Statistics version 21.0 (IBM SPSS Inc., Chicago, IL, USA). Continuous data were summarized with the median (range) or means (± SD). The categorical variables were expressed as a frequency and percentage. Associations between the categorical variables were performed by a Chi-square test. The median OS was estimated using the Kaplan–Meier analysis and the curves were compared by the log-rank test. Multivariate regression analyses were performed using the Cox proportional hazards model. First, the univariate analyses were performed to determine the factors associated with the survival of HCC patients. Then, all variables with a *p*-value < 0.25 in the univariate analysis were included in the multivariable model. We used the backward elimination (LR) as a variable selection method. The proportional hazard ratio (HR) with the 95% confidence intervals (95% CI) were determined both in the univariate and multivariate analyses. A *p*-value < 0.05 was considered to be statistically significant.

## 3. Results

### 3.1. Socio-Demographic Characteristics

A total of 369 patients were included in our study, with males accounting for 67% (Table 1). Based on self-reporting, one third of the patients had a history of alcohol use, while only 10% were smokers. Nearly 75% of patients had an income of ≤5000 Ethiopian Birr (ETB) per month (equivalent to ~95.0 USD).

### 3.2. Clinicopathological Features

The clinicopathological characteristics of the HCC patients are presented in Table 2. The mean BMI was 18.9 ± 3.19 kg/m^2^ and about half of the study’s participants were underweight. According to the Eastern Cooperative Oncology Group (ECOG) performance status criteria, 17 (5%), 115 (33%), 135 (39%), 71 (20%), and 12 (3%) patients were designated with status 0, 1, 2, 3, and 4, respectively. Over 27% of HCC patients had co-morbidities, and hypertension (32%) and diabetes (15%) were among the common co-morbid conditions.

The common clinical symptoms identified were abdominal pain (94%), ascites (53%), and anorexia (47%) (Table 2). A significant proportion of the patients had cirrhosis (58%), and almost all of them (96.7%) were suffering from decompensated liver injuries. Nearly 50% of the HCC patients had viral hepatitis. Hepatitis B virus (HBV) was the most common viral hepatitis (71%) diagnosed in the study population, while hepatitis C virus (HCV) accounted for 28%. However, only a few of the patients (HBV+: *n* = 34, 29% and HCV+: *n* = 3, 7%) received an antiviral treatment. The majority of patients (80%) had multiple and infiltrative types of tumors, and the tumors’ size in 87% of the patients was >5 cm (Range 1.6–23.0 cm). Forty three percent of the patients had metastatic disease, and the lung was the most common site of metastasis (54%). Over 60% of patients had serum AFP levels below 400 ng/mL. Moreover, a liver mass/lesion (45%) and portal vein thrombosis (29%) were among the frequently detected radiological imaging features.

The distribution of stages of HCC is presented in Table 2, as classified by the Barcelona Clinic Liver Cancer (BCLC) staging system. Over 80% of patients (*n* = 263) had a combination of advanced (BCLC-C) and terminal (BCLC-D) stage HCC. The stages of HCC were significantly associated with the baseline ECOG performance score (χ^2^ = 392.3, *p* < 0.001) and the tumor size (χ^2^ = 22.0, *p* < 0.001).

### 3.3. Treatment Modality

More than 75% of patients with advanced (BCLC-C) and terminal (BCLC-D) stage HCC were given supportive care (Table 3). A surgical resection was performed only on 25 (8%) patients, and the majority of patients were with the BCLC-A (*n* = 7) and BCLC-B (*n* = 13) stage HCC. None of the patients in the BCLC-D group underwent a surgical intervention, nor received the transarterial chemoembolization (TACE). About 16% of patients received sorafenib, and only a few patients (*n* = 8) were given TACE therapy. The choice of a treatment modality was strongly associated with the stage of the cancer (*χ^2^* = 145.8, *p* < 0.001).

### 3.4. Survival Outcome

#### 3.4.1. Overall Survival (OS)

The OS of HCC patients was estimated to be 141 days (95% CI: 117–165), as depicted in Figure 2. Based on the Kaplan–Meier survival analysis, the 1-year and 3-year survival probabilities for the entire cohort were 26% and 8%, respectively. The age and sex of the patients were not significantly correlated with the survival outcome (Table 4).

#### 3.4.2. Survival Predictors

The risk of death was increased by about two-fold among underweight patients compared to patients with a normal body weight (HR: 1.81, 95% CI: 1.09–3.01; *p* = 0.021). The difference in the median OS between patients with an AFP value ≤ 400 ng/mL (148 days, 95% CI: 118–178) and >400 ng/mL (118 days, 95% CI: 81–155) was insignificant (*p* = 0.372). Only a few patients, (HBV = 34) and (HCV = 3), received antiviral therapy, and there was no significant survival difference between the patients infected with HBV versus HCV (*p* = 0.170). In contrast to the HCV+ cases, there was statistically significant differences in survival among HBV+ patients with or without antiviral therapy (treated: 469 days, 95% CI: 125–813 vs. untreated: 104 days, 95% CI: 73–135) (Figure 3A). In other words, patients with untreated HBV were 3.5 times more likely to die of HCC compared to the treated HBV group (HR: 3.49; 95% CI: 1.90–6.40; *p* < 0.001). In multivariate analysis, having received a treatment for an HBV infection remarkably decreased the odds of dying by about 67% (HR: 0.33; 95% CI: 0.18–0.61; *p* < 0.001), regardless of the treatment modalities for HCC.

Compared with non-cirrhotic patients (194 days, 95% CI: 109–279), cirrhotic patients had a shorter survival time (104 days, 95% CI: 72–136) (*p* = 0.011) (Figure 3B). In the univariate analysis, patients with cirrhosis showed a higher chance (38%) of dying with HCC compared to patients without cirrhosis (HR: 1.38; 95% CI: 1.07–1.77, *p* = 0.012), but not in the multivariate analysis (Table 4). A significant survival difference (*p* = 0.008) was also observed between patients with a tumor size ≤ 5 cm (243 days, 95% CI: 31–455) and with >5 cm (135 days, 95% CI: 105–165) (Figure 3C). The risk of death was increased by over 6% for every 1 cm increase in the tumor size (*p* < 0.001) (Table 4). Figure 3D demonstrates that the patients with a single tumor survived longer (205 days, 95% CI: 96–314) compared to those with multiple (155 days, 95% CI: 105–205) or infiltrative tumors (118 days, 95% CI: 67–169) (*p* = 0.03). The risk of death was higher in patients diagnosed with infiltrative tumors compared to those with a single tumor type (HR: 1.82, 95% CI: 1.16–2.85; *p* = 0.009).

The median survival time of the patients with intermediate (BCLC-B) (347 days, 95% CI: 247–447), advanced (BCLC-C) (110 days, 95% CI: 69–151), and terminal (BCLC-D) stage HCC (94 days, 95% CI: 70–118) were statistically significantly different (χ^2^ = 39.8, *p* < 0.001). In the Kaplan–Meier curve shown in Figure 4A, notice that the survival probability of patients with an early (BCLC-A) stage HCC was greater than 50%, so their median survival could not be estimated. The subgroup analysis of the BCLC stages revealed that the risk of death for patients with BCLC-D was 12 times higher (HR: 11.72; 95% CI: 2.85–48.26, *p* = 0.001) than patients with an early stage (BCLC-A) HCC. In the multivariate analysis, the chances of dying with HCC was increased by about 46% (HR: 1.46; 95% CI: 1.08–1.96, *p* = 0.013) with each increase in the BCLC stage (Table 4). Similarly, the baseline ECOG performance status was strongly associated with the patients’ survival. In the univariate analysis, the risk of death was increased by about 39% with every unit increase in the ECOG performance (95% CI, 22% to 57%, *p* < 0.001).

The survival outcome was significantly associated with the type of treatment given to HCC patients (*p* < 0.001). Comparing the systemic therapies in Figure 4B, TACE resulted in a longer survival benefit (568 days, 95% CI: 106–1029) over sorafenib (202 days, 95% CI: 145–259) and chemotherapy (99 days, 95% CI: 38–160). Patients who received just supportive care had a median OS of 111 days (95% CI: 87–135). As shown in Figure 5A, patients who underwent surgery had a 52% probability of surviving for a longer period (~565 days) than those without surgery (120 days) (*p* < 0.001). Figure 5B demonstrated that patients with metastasis had a worse survival outcome compared to those with no metastasis (89 days, 95% CI: 56–122 versus 173 days, 95% CI: 139–207; *p* < 0.001).

As illustrated in Table 4, the following variables were included in the multivariate regression model: the stage of HCC (BCLC), cirrhosis, metastasis, the treatment against HBV, and treatment modalities for HCC (excluding palliative care). In the univariate analysis, the tumors’ size, its types, and the patient’s baseline performance (ECOG score) were among the statistically significant survival predictors. However, we did not include them in the multivariable model for the reason that they were all taken into account when the BCLC staging was made. Rather, we assumed that the combined prognostic importance of the tumors’ size/types and ECOG status could be assessed by taking the BCLC stage as a broader (more inclusive) variable. Accordingly, two independent prognostic factors were identified in the final multivariate model: a treatment against HBV (HR: 0.42; 95% CI: 0.22–0.79, *p* = 0.007) and treatment modalities for HCC (HR: 0.67; 95% CI: 0.53–0.86, *p* = 0.001). This indicates that the risk of death decreased by more than 30% in patients having received a treatment for HCC. Given two assumptions, that (1) the access to treatment may be varied across oncology centers and (2) most patients had only received palliative care, we wanted to exclude the HCC treatment modalities from the multivariate analysis and explore the effect of other covariates on the patients’ survival. Thus, when a treatment was removed from the model, the BCLC stage became a significant survival predictor (HR: 1.46; 95% CI: 1.08–1.96, *p* = 0.013) along with the treatment for HBV (HR: 0.33; 95% CI: 0.18–0.61, *p* < 0.001) (Table 4).

## 4. Discussion

The present study included 369 adult patients with confirmed HCC diagnosed between 2016 and 2021 at four oncology care centers in Ethiopia. The median overall survival (OS) of the patients was low. We observed that the majority of HCC patients had multiple and infiltrative types of tumors and with frequent metastasis to the lungs and lymph nodes, so they were presented with advanced stage HCC. Consequently, most of these patients were given palliative care to minimize their pain and improve their quality of life. Very few patients received sorafenib and transarterial chemoembolization (TACE) therapy. A hepatitis B virus (HBV) infection and cirrhosis were identified as the main risk factors for HCC. Only 22% of HBV-positive patients received an antiviral treatment. The stage of HCC and treatment against HBV were the most important survival predictors.

Epidemiological evidences show that HBV is the major etiological agent of viral hepatitis in Africa, the sub-Saharan region in particular [4]. Similarly, our finding revealed an HBV infection as one of the predominant risk factors linked to HCC. However, in Europe, North America, and Asia–Pacific areas, Hepatitis C virus (HCV) is a common risk factor for HCC [12,13]. In the Section 6, we have specifically discussed the viral hepatitis status in Ethiopia and the best way forward. Another well-established risk factor associated with HCC is cirrhosis, which was presented in 58% of HCC patients in this study. Our result is comparable to the report by Jasirwan and colleagues [14] which indicated that nearly 55% of Indonesian HCC patients had underlying cirrhosis. However, the prevalence is low compared to what is globally reported (80–90%) [15]. The discrepancy might be explained by the under-diagnosis of cirrhosis and unidentified risk factors, such as aflatoxin exposure and non-alcoholic fatty liver disease (NAFLD). In Ethiopia, despite the high prevalence of chronic liver diseases including cirrhosis & their risk factors, screening practice for cirrhosis is very poor [16]. One more important etiological factor associated with a chronic liver injury and HCC is alcohol [4]. A third of the HCC patients in our study consumed alcohol; however, we could not verify its relative risk as we could not assess the frequency and amount of the alcohol intake. Thus, a further investigation is warranted. While retrospective studies are informative, their limitations towards a causality inference should be noted.

The economic status of Ethiopian HCC patients is an important issue to better understand the disease and treatment conditions. The majority of our study participants earned <100 USD per month, which is below the World Bank estimate of an average individual monthly income (~200 USD) for low-income countries [7]. The economic condition may further be complicated by the COVID-19 pandemic. With this limited budget, patients have to buy foods, rent a house, as well as cover the cost of their HCC treatment, which poses huge economic constraints. In the present study, about 18% of HCC patients discontinued their treatment due to financial difficulties, which adversely affected their survival outcome. Zou and colleagues [17] indicate that the disease itself poses a huge economic burden for patients, impeding their adherence to medications and other clinical advices. Although the economic consequences of HCC in Ethiopia needs further exploration, reports show that financial costs for HCC hospitalization, especially related to the diseases’ progression, increased substantially in the past decades [17,18].

Serum AFP is often used as a diagnostic marker for HCC, and 400 ng/mL is commonly accepted as a threshold level [19]. In our study, a larger proportion (over 60%) of patients exhibited serum AFP levels <400 ng/mL, suggesting that serum AFP alone cannot be used as a sensitive biomarker for an HCC diagnosis. The diagnostic performance of AFP and its optimal cut-off values for HCC screening are still an issue of controversy [19,20]; thus, potential diagnostic biomarkers with an improved sensitivity, such as those which include extracellular vesicles [21], should further be explored.

The median OS for the entire cohort was estimated to be 141 days (~4.7 months). This finding is in line with the average survival time reported from sub-Saharan countries (~4 months) [4] as well as from China (~5 months) [22]. The 1-year and 3-year survival probabilities in the present study were 26% and 8%, respectively. However, a study from Egypt in HCC patients showed an OS of 15 months, a 1-year survival of 56%, and a 3-year survival of 25% [23]. In their research, only 20% and 24% of the patients, respectively, had advanced (BCLC-C) and terminal (BCLC-D) stage diseases, while 80% of our patients had a combination of the two. The lack of proper treatments in our settings may also contribute to a poor prognosis. Moreover, in our study, underweight patients (BMI < 18.5) had a two-fold increase in the risk of death compared to patients whose BMI > 18.5, as supported by the report of Li and colleagues [22]. This might be linked to anorexia (a loss of appetite) and vomiting, the common clinical presentations at diagnosis. Furthermore, the tumors’ size and metastasis are strongly associated with the survival outcome of patients with HCC. The OS of patients with a larger tumor size (>5 cm) was low compared to those with a smaller size (≤5 cm), and the risk of death increased by over 6% for every 1 cm increase in the tumor size. Several studies have also revealed a decreased survival of HCC patients with an increasing tumor size [24,25,26], which are consistent with our findings.

Our results indicate that untreated HBV patients were 3.5 times more likely to die of HCC compared to treated patients. The multivariate analysis also highlighted the impact of an HBV treatment on the prognosis of HCC patients, as it lowered the chances of dying with HCC by 67% regardless of the type of HCC-related treatment. Previous studies demonstrated that an antiviral therapy for an HBV infection prolongs the OS in HCC patients [27,28]. Moreover, a study shows that the early initiation of an HBV treatment has an effect on reducing the mortality of HCC patients by decreasing the viral load and preventing the progression of hepatitis to cirrhosis [28]. In Ethiopia, the current standard of treatment for a chronic HBV infection is tenofovir if the following criteria are met: an ongoing liver injury evidenced by raised liver enzymes or cirrhosis, or if the patient has a family history of HCC. The treatment can be given for up to 5 years. However, for a chronic HCV infection (the virus should be detected at the RNA level), a highly efficacious (>95% cure rate) direct-acting antiviral (DAA) therapy (combination of sofosbuvir/velpatasvir) is indicated for 3 months. Some patients with viral hepatitis (e.g., HBV) started on an antiviral treatment and followed up with the gastroenterologist long before their diagnosis of HCC. However, mainly due to accessibility and affordability, the patients could not adhere to their medications and thus discontinued the use of the, which resulted in the rapid progression of the disease. Therefore, the early initiation of and adherence to an HBV treatment may reduce the risk of HCC and improve the patients’ chance of survival. Wang et al. [27] suggested that antiviral therapy against hepatitis should also be part of the HCC treatment regimen in order to improve the prognosis. On the other hand, an Editorial paper published in the American Society of Clinical Oncology in 2022 has disclosed a strong correlation between an untreated HBV infection and the risk of extrahepatic cancers, including gastric, colorectal, kidney, and breast cancers [29].

The OS was significantly different between patients with or without cirrhosis. Cirrhotic patients were 38% more likely to die of HCC than non-cirrhotic. Most cirrhotic patients (96.7%) suffered from decompensated liver insults, which possesses a high risk of liver failure and thus a low prognosis. On the other hand, patients with non-cirrhotic liver have a chance to get a surgical resection, which could improve their prognosis [30]. Furthermore, there were differences in the number of cirrhotic patients recruited from different oncology hospitals and this could be one reason for the differences in the survival observed among the study centers (Appendix A). An access to an HCC treatment could also be limited and differed across oncology centers depending on their service capacity and experience, which might result in survival differences.

We have seen statistically significant survival differences among patients with a different ECOG performance status (*p* < 0.001), and mortality risk was increased by nearly 40% for every score increase in the ECOG scale. In present study, more than 60% of the patients had an ECOG score of 2 and above, and such a deteriorated baseline condition may compromise the survival outcome. Consistent with our results, Hsu et al. [31] reported the baseline performance status as an important factor affecting the survival of HCC patients. Since the baseline performance status is one of the key elements in BCLC staging, the stage of HCC is also expected to have an influence on the survival. We showed a statistically significant difference in OS among patients with BCLC-B (347 days), BCLC-C (110 days), and BCLC-D stage HCC (94 days). In line with previous studies [23,24], our result demonstrated that the survival of HCC patients became worse as the disease stage advanced, underscoring the need for an earlier diagnosis and treatment. In multivariate analysis, when the HCC treatment modalities were deliberately excluded from the model, the BCLC stages became the independent survival predictors (HR: 1.46, *p* = 0.013), meaning that the risk of death would have increased by 46% with every unit increase in the BCLC stage if the treatment would have not been given. With the HCC treatment in, the mortality risk estimated for the BCLC stage was around 23%, though it was not statistically significant (HR: 1.23, *p* = 0.206), suggesting modest contributions of HCC-related treatments towards the patients’ survival.

The survival outcome was associated with the type of treatment given to HCC patients. Systemic therapies were linked to a survival advantage compared to palliative care. Patients who had a surgical resection and TACE therapy survived for a relatively long time (median OS: >560 days) compared to patients who received other treatment modalities. The survival benefit of a surgical resection for patients with HCC has been reported in the literature [32,33]. A previous single-center study in Ethiopia revealed a median survival of 351 days for patients who were treated with TACE [34]. The discrepancies may arise from the inherent heterogeneity of patients and differences in the follow-up period (5 vs. 2 years). In multivariate analysis, treatment modalities (excluding supportive care) were identified as an independent prognostic factors for HCC patients, and the mortality risk was decreased by 33% (HR: 0.67, *p* = 0.001). In fact, in our study, the overall effect of an HCC treatment on the survival may be overestimated due to the significantly high contribution of surgery and TACE, in which only 10% of the study’s participants were involved. Therefore, care should be taken when interpreting the effects of a treatment on the patients’ survival, and it should be further explored in future large-scale and possibly prospective studies.

In our research, the exact causes of death in HCC patients could not be verified independently from the patients’ records. However, we anticipated that most of the patients died from the progression of the cancer and/or complications of cirrhosis as many of them were diagnosed with advanced and end-stage HCC and received only supportive care. Yet, some deaths might also be due to non-cancer related illnesses. For example, as a result of lung metastasis, COVID-19 could be the possible cause of death for HCC patients in the last couple of years; however, this may require further investigations and a subgroup analysis.

A major limitation in this retrospective cohort study is missing data (incomplete medical records), which might have introduced selection and recall biases as well as random error during the survival analyses. In addition, only a small number of patients had taken systemic therapies or received a surgical resection, which makes it difficult to compare our findings with other studies. Moreover, the information related to treatment cycles are unavailable and/or some patients discontinued the treatment, which might limit the generalizability of this study on the effects of the treatment. Lastly, the BCLC stage of HCC for some patients was determined, not at the time of diagnosis but during the data extraction through reviewing their clinical profile; so, a subjective assessment bias could exist. Despite these limitations, there are findings of real interest in the prevention and treatment of HCC in Ethiopia in our study. Our results may also provide data for the establishment of a nation-wide cancer registry, accounting for a diverse patient population recruited from different cancer treatment centers.

## 5. Conclusions

The overall survival of HCC patients in Ethiopia is low, as most of the patients are presented with the advanced stage of the disease and were given only palliative care. The hepatitis B virus (HBV) and cirrhosis are important risk factors which are associated with HCC. The stage of HCC and the treatment against HBV are the main independent survival predictors.

## 6. Current Challenges and Future Prospects

Viral hepatitis is a significant public health problem in developing nations, including Ethiopia, and vaccination is the most effective strategy in preventing an HBV infection. Vaccination coverage varies significantly across countries. In European countries, the vaccination coverage is over 90% [35], while the coverage in Africa, even among healthcare workers, has been reported to be about 25% [36]. Shockingly, in sub-Saharan Africa, the coverage is estimated to be only 6%, and this is mainly due to the limited availability and affordability of vaccines. In recent years, Ethiopia has introduced a hepatitis B vaccine into the routine infant immunization program, which is a positive milestone. However, studies show that there are low levels of antibody response to the vaccine; hence, after the vaccination, the antibody response against HBV should be carefully monitored. Another challenge in combating viral hepatitis in Africa is the high cost of antiviral medications, such as tenofovir and entecavir. On the other hand, for the treatment of HIV-infected patients, tenofovir is freely available in public hospitals through donations. Therefore, if funding agencies authorize and HIV care units are well-coordinated, tenofovir can also be used for the treatment of viral hepatitis, because HBV is claiming more deaths than HIV [37]. Moreover, collaborating with the HIV divisions in creating awareness on high-risk sexual behavior in the community is key to controlling the transmission of viral hepatitis as well. As discussed above, patients who have chronic viral hepatitis are at an increased risk of developing HCC; hence, strengthening the prevention and control efforts against HBV would reduce the prevalence of HCC. According to the WHO [38], the incidence and mortality rates of HCC in Ethiopia are projected to double by 2040, and viral hepatitis is suggested as one of the potential contributing factors. This body of evidence highlights the need for a concerted effort by health professionals, government policymakers, funding agencies, as well as community leaders in promoting HCC screening and surveillance programs, in the prevention and control of viral hepatitis, and in building the capacity of primary healthcare providers, which ultimately can improve the survival of patients with HCC.

Though it is beyond the scope of this paper to discuss cancer staging systems, we would like to comment on the HCC staging practices in Ethiopia. From our observation, oncology centers are staging HCC using different systems, resulting in heterogeneous staging information. During the data collection, we asked oncology residents to re-stage the disease based on the available clinical and pathological information following the Barcelona Clinic Liver Cancer (BCLC) staging technique. Only then were we able to combine the datasets coming from different study centers (data pooling). We know that cancer staging is a cornerstone of patient care as it helps clinicians to select the best treatment regimen and predict the patients’ prognosis. Besides the BCLC staging system, there are also other staging methods, including the tumor–node–metastasis (TNM) and the Okuda staging system. Because of heterogeneity of HCC, no single staging system is accepted globally, however, we would like to advise clinicians in Ethiopia to implement a uniform kind of staging system that would facilitate the patient’s referral process and improve the oncology care services.

## Figures and Tables

**Figure 1 cancers-15-00193-f001:**
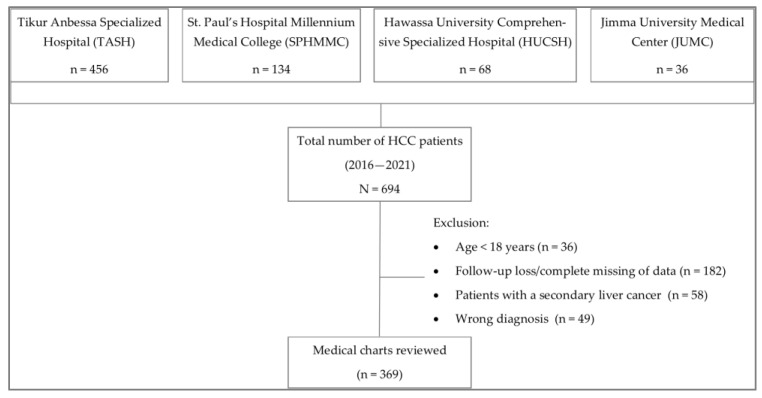
Flow chart showing the number of HCC patients enrolled in the study.

**Figure 2 cancers-15-00193-f002:**
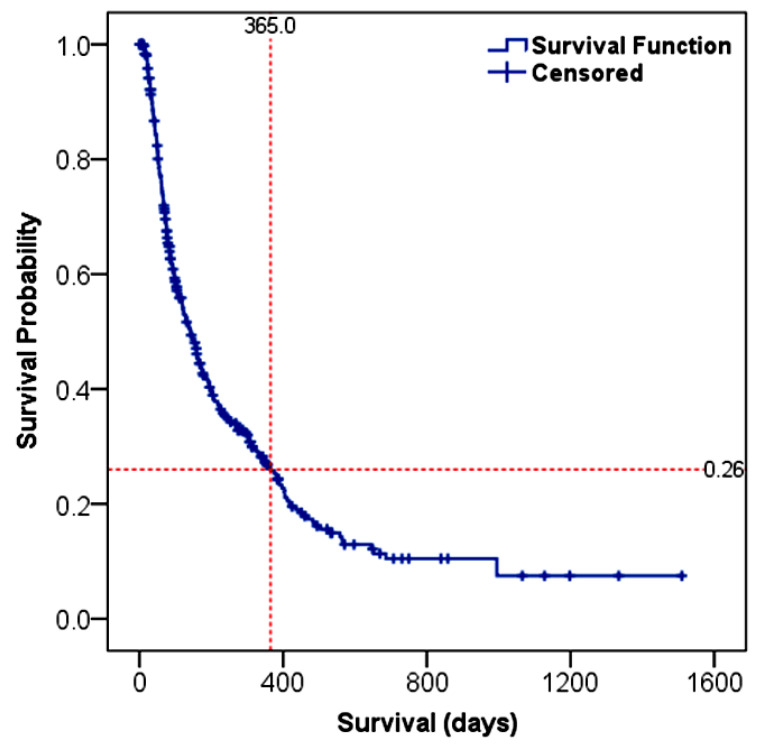
Kaplan–Meier survival curve for Ethiopian HCC patients. The red dotted-lines indicate the 1-year survival probability. Censored subjects are designated as “+” mark and represent those patients who were lost during follow-up or alive at the end of the study.

**Figure 3 cancers-15-00193-f003:**
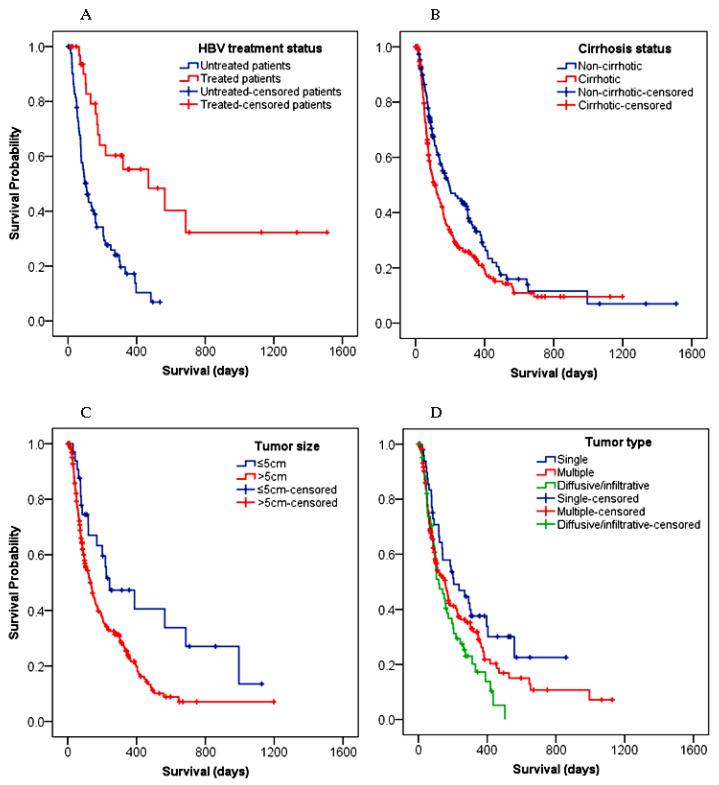
Survival differences among HCC patients: (**A**) HBV treated and untreated; (**B**) cirrhotic versus non-cirrhotic; (**C**) with tumor sizes of ≤5 cm versus >5 cm; (**D**) with different tumor types. Censored patients are indicated as the mark “+” and represent patients who were lost during follow-up or alive at the end of the study. HBV, hepatitis B virus.

**Figure 4 cancers-15-00193-f004:**
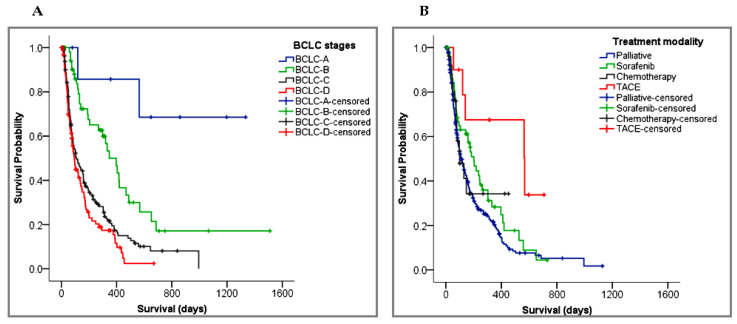
Kaplan–Meier survival curves for HCC patients with different: (**A**) BCLC stages; (**B**) treatment modalities. Censored patients are marked as “+” and represent patients who were lost during follow-up or alive at the end of the study. BCLC, Barcelona Clinic Liver Cancer staging system. TACE, transarterial chemoembolization.

**Figure 5 cancers-15-00193-f005:**
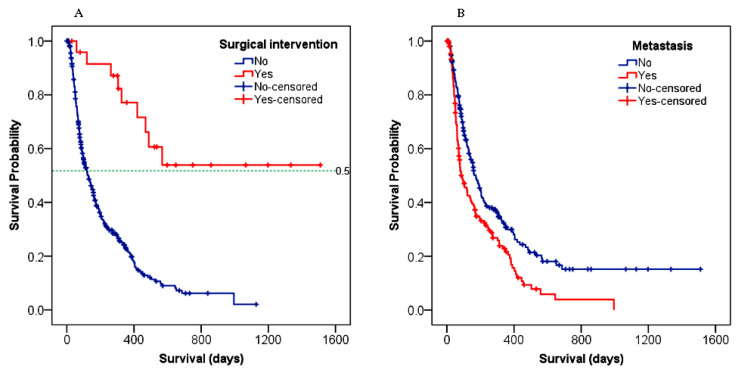
Survival differences among HCC patients: (**A**) with or without metastasis; (**B**) who received surgical intervention or not. “+” symbol represents censored patients who failed to follow-up or were alive at the end of the study.

**Table 1 cancers-15-00193-t001:** Socio-demographic characteristics of HCC patients.

Characteristics	HCC Patients, N (%) ^1^
Age, years (mean ± SD)	52.0 ± 15.56
Sex Male Female	246 (67) 123 (33)
Educational status (*n* = 327) None ^2^ Primary Secondary College/University	133 (41) 74 (22) 68 (21) 52 (16)
Economic status (monthly income, ETB ^3^) (*n* = 288) <500 501–1000 1001–5000 5001–10,000 >10,000	127 (44) 8 (3) 79 (27) 63 (22) 11 (4)
Alcohol use (*n* = 366), yes	123 (37)
Smoking status (*n* = 366), yes	38 (10)
Physical exercise ^4^ (*n* = 325), no	314 (97)

^1^ Values are expressed as frequency (%) unless and otherwise specified. ^2^ Lack of formal education. ^3^ ETB–Ethiopian Birr (1 USD = ~53 ETB). ^4^ For the purpose of this study, physical exercise was defined as moderate to intensive sport activities.

**Table 2 cancers-15-00193-t002:** Clinicopathological characteristics of HCC patients.

Variables	Frequency, N (%)
**BMI ^1^ (*n* = 96)** Underweight (<18.5 kg/m^2^) Normal weight (18.5–24.9 kg/m^2^) Overweight (≥25 kg/m^2^)	47 (49) 45 (47) 4 (4)
**Comorbidities (*n* = 368)** Yes No	100 (27) 268 (73)
**Performance score ^2^ (*n* = 350)** ECOG 0 ECOG 1 ECOG 2 ECOG 3 ECOG 4	17 (5) 115 (33) 135 (39) 71 (20) 12 (3)
**BCLC stage ^3^ (*n* = 326)** BCLC-A BCLC-B BCLC-C BCLC-D	8 (2) 55 (17) 166 (51) 97 (30)
**Sign and symptoms ^4^** Abdominal pain Ascites Anorexia Weight loss Jaundice	344 (94) 195 (53) 171 (47) 141 (38) 56 (15)
**Degree of pain (*n* = 340)** Mild Moderate Severe	139 (41) 178 (52) 23 (7)
**History of cirrhosis (*n* = 369)** Present Absent	213 (58) 156 (42)
**Cirrhosis (*n* = 213)** Compensated Decompensated	7 (3) 206 (97)
**Viral hepatitis (*n* = 355)** Positive Negative	165 (47) 190 (53)
**Type of viral hepatitis ^5^ (*n* = 165)** HBV HCV HBV + HCV	117 (71) 46 (28) 2 (1)
**Type of tumor (*n* = 266)** Single Multiple Diffuse/infiltrative	50 (19) 152 (57) 64 (24)
**Tumor size (*n* = 262)** ≤5 cm >5 cm	34 (13) 228 (87)
**Metastasis (*n* = 366)** Yes No	157 (43) 209 (57)
**Liver function estimates ^6^** ALT, U/L, median [IQR] AST, U/L, median [IQR] ALP, U/L, median [IQR] Total bilirubin, mg/dL, median [IQR] Albumin, g/dL, mean ± SD	43.0 [26–80] 82.0 [44–146.8] 203.5 [129–326.3] 1.0 [0.6–1.7] 3.4 ± 0.74
**AFP ^7^ (*n* = 295)** ≤400 ng/mL >400 ng/mL	185 (63) 110 (37)

Values are presented as frequency (%) unless specified. ^1^ BMI–body mass index WHO classification. ^2^ Eastern Cooperative Oncology Group (ECOG) Performance Status scoring system. Zero–fully active, able to carry out all pre-disease function without restriction; 1–restricted in physically strenuous activity but ambulatory and able to carry out work of a light or sedentary nature (e.g., light house work, office work); 2–ambulatory and capable of all self-care but unable to carry out any work activities (up and about 50% of waking hours); 3–capable of only limited selfcare, confined to bed or chair more than 50% of waking hours; and 4–completely disabled, cannot carry on any selfcare, totally confined to bed or chair. ^3^ BCLC, Barcelona Clinic Liver Cancer staging system adopted from the National Comprehensive Cancer Network (NCCN) guideline [10]. ^4^ A patient could present with multiple symptoms. ^5^ HBV–hepatitis B virus; HCV–hepatitis C virus. ^6^ ALT, alanine aminotransferase; AST, aspartate aminotransferase; and ALP, alkaline phosphatase. ^7^ AFP, alpha-fetoprotein, IQR–Interquartile range.

**Table 3 cancers-15-00193-t003:** Types of treatment given to HCC patients with respect to the BCLC stages of HCC.

BCLC Stage	Treatment Given (*n* = 324)				
	Surgery	TACE	Sorafenib	Chemotherapy ^1^	Palliative
BCLC-A (Early)	7	1	0	0	0
BCLC-B (Intermediate)	13	5	9	1	26
BCLC-C (Advanced)	5	2	36	12	110
BCLC-D (Terminal)	0	0	6	2	89

^1^ Chemotherapy includes: 5-fluorouracil + leucovorin + oxaliplatin (FOLFOX); cisplatin + doxorubicin; and cisplatin + gemcitabine. BCLC, Barcelona Clinic Liver Cancer staging system. TACE, transarterial chemoembolization.

**Table 4 cancers-15-00193-t004:** Univariate and multivariate analysis of factors associated with survival of HCC patients.

Covariates	Univariate Analysis		Multivariate Analysis (with HCC Treatment)		Multivariate Analysis (without HCC Treatment)	
	HR (95% CI)	*p*-Value	HR (95% CI)	*p*-Value	HR (95% CI)	*p*-Value
Age (years)	1.00 (0.99–1.01)	0.744	-	-	-	-
Gender	1.00 (0.77–1.29)	0.986	-	-	-	-
Tumor size (cm)	1.06 (1.03–1.10)	<0.001 **	-	-	-	-
ECOG Score	1.39 (1.22–1.57)	<0.001 **	-	-	-	-
BCLC stage	1.72 (1.44–2.04)	<0.001 **	1.23 (0.89–1,69)	0.206	1.46 (1.08–1.96)	0.013 *
Viral hepatitis ^1^	0.98 (0.77–1.26)	0.897	-	-	-	-
HBV treatment	0.29 (0.16–0.53)	<0.001 **	0.42 (0.22–0.79)	0.007 *	0.33 (0.18–0.61)	<0.001 **
HCV treatment	0.26 (0.03–1.89)	0.181	-	-	-	-
Metastasis	1.58 (1.24–2.02)	<0.001 **	0.94 (0.59–1.50)	0.793	1.05 (0.66–1.65)	0.851
Cirrhosis	1.38 (1.07–1.77)	0.012 *	1.45 (0.93–2.27)	0.103	1.51 (0.97–2.35)	0.069
AFP (ng/mL)	1.00 (1.00–1.00)	0.934	-	-	-	-
HCC treatment ^2^	0.51 (0.38–0.67)	<0.001 **	0.67 (0.53–0.86)	0.001 *	-	-

Factors: ^1^ viral hepatitis test–positive vs. negative; ^2^ excluding palliative care; * represent *p* < 0.05; ** represent *p* < 0.001. ECOG, Eastern Cooperative Oncology Group; BCLC, Barcelona Clinic Liver Cancer staging system; HBV, hepatitis B virus; HCV, hepatitis C virus; AFP, alpha-fetoprotein, HR, hazard ratio; CI, confidence interval.

## Data Availability

The data that support the findings of this research are available upon reasonable request.

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
