# Peer review of "Clinicopathological Features and Survival of Patients with Hepatocellular Carcinoma in Ethiopia: A Multicenter Study"

_cancers, 2022, doi:10.3390/cancers15010193_

Round 1

Reviewer 1 Report

Authors analysed clinicopathological features and survival of patients with HCC in Ethiopia.

The introduction provides sufficient background.    

In the methods the authors should clarify:

1)        If the radiologic diagnosis of HCC was made with contrast-enhanced CT/MRI or CEUS?

2)        If in all patients without cirrhosis, the diagnosis of HCC was confirmed by a histological examination as request in guidelines?

3)        On which criteria was based the diagnosis of cirrhosis? 

4)        Who chose the management of HCC: oncologist/surgeon/hepatologist or multidisciplinary team?

5)        Patients with viral hepatitis begun the treatment before or after the diagnosis of HCC?

In the results the authors should give information about the stage of cirrhosis in cirrhotic patients (compensated/decompensated/Child Pugh score) and should clarify the cause of dead in patients who not-survive advance HCC, complication of cirrhosis or non-hepatic cause of death.

In the discussion authors should explain why the percentage with HCC on cirrhotic liver was only 37% considering that the percentage in medical literature is about 80-90%.

Author Response

Dear reviewer,

We thank you very much for your insightful remarks on our paper. We have taken the opportunity to look back at our original data and re-analyzed them thoroughly, which resulted major changes and improvement on our findings. Therefore, we truly appreciate the time and effort you dedicated to providing us such a constructive feedback. We have attempted to address all the comments in a point-by-point manner, and changes have been made in the manuscript accordingly.

Sincerely,

Reviewer 2 Report

I read with pleasure this paper reporting outcomes of HCC patients in Ethiopia. The Authors reported the results of a retrospective multicenter study involving the 4 largest hepatology centres in the Country. The recruited cohort consisted mostly of patients with a symptomatic diagnosis of HCC, with poor performance status and untreated viral hepatitis. As a result, the survival rates were very low. This paper is of global interest as reports from Ethiopia and SubSaharian Africa, in general, are scant and based on the monocenter series. The results help to understand which public health measures could be introduced to reduce the burden of disease. There are some major points needing attention:

1)      Paragraph 3.2 (Clinical features): The current categorization of patients according to the BCLC classification is wrong. Patients with ECOG-PS>1 should be categorized as terminal HCC (BCLC-D). Yet, the authors reported and analyzed very early, early, intermediate, and advanced HCC. This distinction is particularly important in a population in which severely symptomatic patients are predominant and ECOG-PS is the main driver of survival.

2)      Paragraph 3.3 (Treatment modalities): Treatment is a key aspect as it is a known determinant of survival. The authors should add a table to this paragraph using BCLC stages as rows and treatments as columns, or vice versa. In this way, the readers will understand which patients were treated and with which modalities easily.

3)      Paragraph 3.4.2 (Survival predictors): It is not clear at all how the multivariable model was designed. Please state which variables were included, and report the HR (95% CI, and p-value) for every variable included, even if statistical significance was not met. Also, I suggest trying a further experiment, i.e build two different models, one including treatment and the other one not including it. Since access to treatment can be very different across centres, the interpretation of the effects of the treatment itself could be tricky.

4)      Paragraph 3.4.2 (Survival predictors): From the last point, did the survival differ according to the recruiting centre? Please include this information in the analyses.

1)      Paragraph 3.4.2 (Survival predictors): The effects of anti-viral treatments should be analysed in HBV+ and HCV+ patients separately. Also, please describe which antiviral treatments were administered. I gather that antiviral treatments consisted mainly of NUCs for HBV. In that case, it would not be appropriate to consider “untreated viral patients” as a whole group.

Author Response

Dear reviewer,

Thank you very much for your insightful comments on our paper, and for your understanding on how relevant our study and the results are in the context of developing nations. We took our time to properly address your critical, but constructive, questions. We have been able to look back at our original data and re-analyzed them thoroughly, which resulted major changes and improvement on our findings. Therefore, we truly appreciate the time and effort you dedicated to reviewing our manuscript. We have attempted to address all the comments in a point-by-point manner, and changes have been made in the manuscript accordingly.

Sincerely,

Round 2

Reviewer 2 Report

I would like to thank the Authors for having considered and addressed all of the points I raised. I feel that the manuscript has improved markedly and think that it will be a nice addition to Cancers.

I have no further comments.